# Marine Plasmalogens: A Gift from the Sea with Benefits for Age-Associated Diseases

**DOI:** 10.3390/molecules28176328

**Published:** 2023-08-29

**Authors:** Shinji Yamashita, Taiki Miyazawa, Ohki Higuchi, Mikio Kinoshita, Teruo Miyazawa

**Affiliations:** 1Department of Life and Food Sciences, Obihiro University of Agriculture and Veterinary Medicine, Obihiro 080-8555, Japan; syamashita@obihiro.ac.jp (S.Y.); kinosita@obihiro.ac.jp (M.K.); 2Food and Biotechnology Platform Promoting Project, New Industry Creation Hatchery Center (NICHe), Tohoku University, Sendai 980-8579, Japan; taiki.miyazawa.b3@tohoku.ac.jp (T.M.); higuchi@hokkaido-bpi.co.jp (O.H.)

**Keywords:** aging, ascidian, dementia, DHA, glycerophospholipids, inflammation, marine resources, MCI, oxidative stress, plasmalogen

## Abstract

Aging increases oxidative and inflammatory stress caused by a reduction in metabolism and clearance, thus leading to the development of age-associated diseases. The quality of our daily diet and exercise is important for the prevention of these diseases. Marine resources contain various valuable nutrients, and unique glycerophospholipid plasmalogens are found abundantly in some marine invertebrates, including ascidians. One of the major classes, the ethanolamine class (PlsEtn), exists in a high ratio to phospholipids in the brain and blood, while decreased levels have been reported in patients with age-associated diseases, including Alzheimer’s disease. Animal studies have shown that the administration of marine PlsEtn prepared from marine invertebrates improved PlsEtn levels in the body and alleviated inflammation. Animal and human studies have reported that marine PlsEtn ameliorates cognitive impairment. In this review, we highlight the biological significance, relationships with age-associated diseases, food functions, and healthcare materials of plasmalogens based on recent knowledge and discuss the contribution of marine plasmalogens to health maintenance in aging.

## 1. Introduction

Aging decreases metabolic, clearance, and immune functions and increases oxidative and inflammatory stress [1,2]. For example, a decrease in the clearance of amyloid-β (Aβ) from the brain by aging is hypothesized to develop into Alzheimer’s disease (AD) via chronic oxidation and inflammation [3]. As human life expectancy is high in contemporary society, it is necessary to reduce the risk of age-associated diseases.

At the beginning of life, the sea is still mysterious, and marine resources possess various valuable nutrients that are not found in land resources. Docosahexaenoic acid (DHA) and eicosapentaenoic acid (EPA) as *n*-3 polyunsaturated fatty acids (PUFAs) contained in marine food contribute to health maintenance in current diets containing excessive *n*-6 PUFAs and saturated fatty acids [4]. Algae-derived xanthophylls, including fucoxanthin and astaxanthin, have been reported to exhibit not only antioxidative activity but also various food functions [5].

Plasmalogens are glycerophospholipids (GPLs) with a vinyl ether bond at the *sn*-1 position. In particular, the ethanolamine class (PlsEtn) is abundantly distributed in the nervous system, including the brain, and PlsEtn is thought to exhibit nerve protection as an endogenous antioxidant. In some marine invertebrates, high concentrations of PlsEtn exist in the muscles and viscera because of lower neural differentiation than that in vertebrates [6,7,8]. Reportedly, PlsEtn levels in the brain and blood are altered by aging and some age-associated diseases, including AD [9,10], and the administration of PlsEtn from marine invertebrates (marine PlsEtn) alleviates the impairments caused by aging [11,12]. In this review, we discuss and summarize the biological significance of plasmalogens and their relationship with age-associated diseases, food functions, and healthcare materials.

## 2. Structures and Roles of Plasmalogens

### 2.1. Structures and Distribution

GPLs are composed of a glycerol bone with two fatty chains and a polar head with phosphate. They are classified by their polar heads. For example, ethanolamine and choline GPLs (EtnGpl and ChoGpl), the major classes of GPLs found in the biological membranes of mammals, possess phosphoethanolamine and phosphocholine as their polar head, respectively. Moreover, GPLs are classified into three subclasses with alkyl, alkenyl, or acyl linkages at the *sn*-1 position of the glycerol moiety. In the case of EtnGpl, the subclasses are 1-*O*-alkyl-2-acyl-*sn*-glycero-3-phosphoethanolamine (PakEtn), 1-*O*-alkenyl-2-acyl-*sn*-glycero-3-phosphoethanolamine (PlsEtn), and 1,2-diacyl-*sn*-glycero-3-phosphoethanolamine (PtdEtn). In the case of ChoGpl, the subclasses are 1-*O*-alkyl-2-acyl-*sn*-glycero-3-phosphocholine (PakCho), 1-*O-*alkenyl-2-acyl-*sn*-glycero-3-phosphocholine (PlsCho), and phosphatidylcholine (PtdCho). The alkenylacyl form is called plasmalogen. Fatty alcohols at the *sn*-1 position of plasmalogens consist primarily of the C16:0 (palmitoyl), C18:0 (stearoyl), or C18:1 (oleoyl) carbon chains, whereas those at the *sn*-2 position consist primarily of PUFAs such as DHA and arachidonic acid (ARA).

In mammals, plasmalogens are ubiquitous in the body (Table 1 and Table 2). However, their ratio to phospholipids depends on the organ. The ratio of PlsEtn to EtnGpl is high in the nervous system, including the brain (50–90%; especially high ratio in the myelin sheath), heart (25–75%), blood plasma (approximately 50%), and red blood cells (RBCs; 45–65%), whereas it is low in the liver (<10%) [13,14,15]. The ratio of PlsCho to ChoGpl is basically lower than that of PlsEtn, with the highest being 5–40% in the heart. Tissues having high levels of plasmalogens (i.e., the brain and heart) indicate low expression of lyso-plasmalogenase, which hydrolyzes the vinyl ether bond of lyso-plasmalogens, while tissues having low levels, like the liver, exhibit high expression [16].

### 2.2. Biological Role

Plasmalogens play various roles in biological membranes. Plasmalogens are closer to the *sn*-1 and *sn*-2 chains than those in diacyl-types; therefore, plasmalogens strengthen lipid membranes and lower their fluidity, both of which are related to cellular functions [56,57]. Plasmalogens scavenge reactive oxygen species (ROS), such as singlet oxygen (^1^O_2_) and superoxide (O_2_^−^), at their alkenyl (vinyl ether) linkages [58,59]. Plasmalogens also store PUFAs at the *sn*-2 position, and the PUFAs are released by phospholipase A_2_ (PLA_2_), including the plasmalogen-selective PLA_2_ [60]. The released PUFAs can be metabolized into eicosanoids and docosanoids, which exhibit various bioactivities, including the promotion and inhibition of inflammation [61]. PLA_2_ isoforms differ in specificity depending on the linkage, length, and unsaturation of the carbon chains at the *sn*-1 and *sn*-2 positions; therefore, plasmalogens and PLA_2_ can control biological reactions [62].

Plasmalogens are synthesized in the peroxisomes and endoplasmic , and the initial reaction is an acylation of dihydroxyacetone phosphate (DHAP) by acyltransferase (DHAP-AT). Peroxisome biogenesis disorders, which are congenital dysfunctions of the peroxisome, impair the metabolism of lipids and ROS (e.g., low plasmalogen levels and promotion of inflammation), thus resulting in various symptoms, including hypomyelination. Zellweger syndrome and rhizomelic chondrodysplasia punctata (RCDP) are severe types of peroxisome biogenesis disorders, and most patients with their diseases die during infancy. Mutant cells deficient in plasmalogen synthesis exhibit low resistance to oxidation, and the addition of alkylglycerol as a plasmalogen precursor increases plasmalogen levels to enhance oxidative resistance [63]. Plasmalogen levels in cells are tightly administered; as feedback inhibition, excessive levels of plasmalogens reduce the levels of fatty acyl-CoA reductase 1 (Far1), which provides fatty alcohols in the synthesis, by promoting the degradation of the protein to control the plasmalogen levels [64]. Cells, mice, and humans deficient in peroxisomes have decreased PlsEtn levels but increased PtdEtn levels alternatively, and the total EtnGpl levels are maintained [65].

The significance of plasmalogens in the brain and nerves has been extensively investigated. Because PlsEtn easily forms an inverse hexagonal phase at body temperature compared with PtdEtn, PlsEtn is involved in membrane fusion during synaptic transmission [66]. In plasmalogen-deficient mice (knockout of the DHAP-AT gene), myelination in the spinal cord and optic nerves was improved by intraperitoneal (i.p.) injection of alkyl glycerol [67]. The knockdown of the DHAP-AT gene in the hippocampus of adult mice decreases memory function and neurogenesis [68]. According to a recent study, a decrease in PlsEtn levels reduces cholesterol synthesis; because the brain does not receive cholesterol from the blood, low PlsEtn levels in the brain lead to a shortage of 24-hydroxycholesterol as a cholesterol metabolite and a ligand of the liver X receptor (LXR), which inhibits myelin formation via LXR [64]. Additionally, plasmalogens are reported to be related to ferroptosis, a type of cell death caused by the accumulation of phospholipid hydroperoxides. Saturated fatty acids, which are metabolized to fatty alcohols by Far1, enhance the effects of ferroptosis inducers [69]. Because Far1 levels receive feedback inhibition by plasmalogens, such as those described above, a decrease in the brain levels of plasmalogens may induce neurocyte death.

In terms of the action of plasmalogens on the heart and blood, there are reports on their antioxidant properties [70,71,72]. In macrophages, which are derived from monocytes and are responsible for the immune response, PlsEtn affects the signal transduction of inflammation by providing ARA as a source of eicosanoids [73] and by controlling the number and size of lipid rafts [54]. Because plasmalogens in the heart and blood plasma possess the same levels of PlsCho as PlsEtn, whereas those in other organs are predominantly PlsEtn, PlsCho may play important roles in the heart and blood [74]. Mice deficient in mitochondrial transacylase tafazzin, a model of Barth syndrome, had decreased cardiac levels of PlsCho but not PlsEtn, whereas a significant decrease in PlsEtn levels was observed in the brain, kidney, and liver [45,75]. However, the biological significance of plasmalogens in the organs remains unclear.

## 3. Alterations of Plasmalogens in Aging and Associated Diseases

### 3.1. Aging

Aging is associated with cellular functional impairments, including mitochondrial dysfunction, that induce chronic oxidative and inflammatory stress [1,2]. For example, in human plasma and RBCs, the levels of GPL peroxides (PLOOHs), the first oxides of GPLs, increase depending on aging, and hyperlipidemia enhances the accumulation of PLOOHs with aging [76,77,78]. Reportedly, ChoGpl peroxides (PCOOHs) evoke monocyte adherence to the arterial wall during the initiation of atherosclerosis [79], and aged RBCs, which accumulate PLOOHs, reduce their ability to transport oxygen to the organs. Therefore, aging synergistically increases the risk of various diseases caused by oxidative and inflammatory stress.

In the human brain, plasmalogen levels peak in middle age and tend to decrease in old age [80,81], along with myelin (rich in PlsEtn) [82]. Free aldehydes and α-hydroxyaldehydes derived from the *sn*-1 position of plasmalogen oxides are markedly increased in the older adult human brain [81]. In human plasma and RBCs, plasmalogen levels decrease with aging [9]. This is because aging reduces peroxisomal functions [83] and promotes the degradation of the vinyl ether bond in plasmalogens by cytochrome *c* and oxidative stress [84]. A decrease in the levels of the antioxidant GPL plasmalogens reduces oxidative resistance and induces membrane dysfunction, including that of lipid rafts.

### 3.2. Alzheimer’s Disease

AD is the most common form of dementia worldwide. One of the pathological characteristics of AD is the progressive aggregation and accumulation of Aβ peptide in the senile plaques of the human brain. Because brain Aβ, particularly its fibril form, is highly neurotoxic, the progressive aggregation of Aβ is a critical step in AD pathogenesis [85]. Aging promotes chronic inflammation and Aβ accumulation, increasing the risk of developing and advancing AD [86,87].

Plasma Aβ is hypothesized to readily contact RBCs and impair their functions in the circulating human blood [88,89]. Our group and other researchers investigated this hypothesis and found that Aβ induces oxidative injury in RBCs by binding to them and causing the accumulation of PLOOHs [90,91]. Additionally, Aβ induces the binding of RBCs to endothelial cells and decreases endothelial viability, possibly by generating oxidative and inflammatory stress [92]. Moreover, RBC Aβ and PCOOH levels increase with age in healthy subjects, and RBC PCOOH levels increase in patients with AD [93,94].

By contrast, the levels of PlsEtn and PlsCho have been reported to decrease specifically in the brain and serum of patients with AD, depending on the severity of the disease [10,50,95,96]. We also analyzed the levels of Aβ, PCOOHs, and PlsEtn in the blood of patients with AD and their spouses (healthy subjects) (Figure 1) [97]. Evidently, plasma from patients with AD exhibited lower concentrations of PlsEtn species, particularly PlsEtn bearing a DHA moiety. Additionally, lower PlsEtn and higher PCOOH levels were observed in the RBCs of patients with AD. In both the AD and control blood samples, RBC PCOOH levels were correlated with plasma levels of Aβ_40_, particularly in patients with AD, and each PlsEtn species exhibited different correlations with plasma Aβ. Furthermore, PlsEtn bearing DHA suppressed the formation and disruption of Aβ fibrils in vitro, whereas the PtdEtn and PtdCho species were not affected (Figure 2) [97]. Together with the toxicities of Aβ fibrils [85,86,87,88,89,90,91,92], a decrease in PlsEtn levels is believed to weaken the blood and brain, thus advancing AD symptoms.

### 3.3. Parkinson’s Disease

Parkinson’s disease (PD) is characterized by motor dysfunction caused by the loss of dopaminergic neurons in the substantia nigra pars compacta and the depletion of dopamine in the nigrostriatal pathway. Aging increases PD morbidity, which is responsible for mitochondrial dysfunction and chronic oxidative and inflammatory stress [98]. Furthermore, in the advanced stages of PD, some patients develop dementia, and Aβ deposition, a pathological feature of AD, is observed in the brain [99]. In the PD brain, plasmalogens were lower than those in the control brain, but not significantly; however, in lipid rafts from the PD brain, the levels of plasmalogens were significantly lower [95,100]. Lipid rafts are strongly associated with signal transduction. It has been suggested that a reduction in plasmalogen levels decreases membrane strength and the levels of eicosanoids and docosanoids from PUFAs, which affect signal transduction. Additionally, PlsEtn levels in the plasma and RBCs of patients with PD are significantly lower than those in control subjects [101,102]. The lipid composition of blood is thought to reflect that of the organs [103]. In contrast, the reduction in blood PlsEtn rich in PUFAs may induce impairment of brain blood flow, which is associated with PD [104]. It would be interesting to determine whether, in patients with PD, blood PlsEtn levels fall earlier than those of brain lipid rafts.

### 3.4. Arteriosclerosis

Aging and obesity induce chronic inflammation and dysfunction of the lipid metabolism, leading to arteriosclerosis and cardiovascular disease [100]. The oxidized low-density lipoprotein (LDL) containing PCOOHs induces epidemic apoptosis [101,102] and promotes monocyte adherence to the arterial walls [79]. Plasma PCOOH levels increase during aging, particularly during hyperlipidemia [76]. By contrast, the addition of plasmalogens in vitro increased the oxidative stability of LDL, which is dependent on its content [71]. Patients with hyperlipidemia, particularly those with high levels of both triglycerides and cholesterol, exhibit lower serum plasmalogen levels than healthy subjects; serum plasmalogen levels are negatively correlated with serum triglyceride levels [103]. Patients with familial hypercholesterolemia indicate lower plasmalogen levels in plasma and LDL compared with healthy subjects; the supplementation of α-tocopherol as a lipophilic antioxidant increases their levels [32]. Serum levels of plasmalogens, particularly PlsCho bearing oleic acid, exhibit correlations with atherosclerosis-related factors (positive with protective functions and negative with promotion factors) [104,105]. Patients with coronary artery disease have lower levels of plasma PlsCho [106,107] and RBC PlsEtn [31]. Animal models of ischemia show a decrease in plasmalogen levels and an increase in lipid oxides, including α-hydroxyaldehyde as a plasmalogen oxide, in the heart and spinal cord [108,109,110]. Together with their antioxidant properties [70,71,72], it is thought that blood plasmalogen levels are easily influenced by oxidative stress and abnormal lipid metabolism, and a decrease in plasmalogen levels reduces resistance to their stress. Therefore, maintaining blood levels may be important for health of the circulatory system.

### 3.5. Cancer

Cancers are accentuated by DNA damage, apoptosis, dysregulation of cell homeostasis, and dedifferentiation [111], which are caused by chronic inflammation from aging as well as a dangerous lifestyle [1,112]. Reportedly, cancers alter the metabolism of GPLs, including plasmalogens [113]. Many studies have reported that higher levels of ether lipids (i.e., plasmalogens and alkyl-GPLs) are observed in cancer cells and tumors than in normal cells and tissues. This is due to the upregulation of biosynthesis enzymes of ether lipids in malignant cells and ether lipids’ contribution to the promotion of cancer progression via the generation of signaling lipids [114]. Hypoxia, which is a factor in cancer progression, increases plasmalogen levels in in vitro colon cancer cells by activating the re-acylation of lyso-plasmalogens, not plasmalogen synthesis [115]. Patients with gastric carcinoma show higher plasma levels of PlsEtn [116]. In contrast, some studies reported that cancers decreased plasmalogen levels, esophageal tumors decreased PlsEtn and other phospholipids [117], and patients with pancreatic cancer exhibited lower serum levels of PlsEtn than control subjects [118]. Lyso-PlsEtn and *n-3* PUFAs, provided from PlsEtn by epidermal secretory PLA_2_ (sPLA_2_-IID), promote skin cancer hypergrowth and reduce antitumor immunity, respectively [119]. Additionally, ferroptosis, an attractive target for anticancer therapy [120], is associated with plasmalogens. Plasmalogens containing PUFAs accelerate ferroptosis via lipid peroxidation [121], while plasmalogens with non-PUFAs suppress ferroptosis enhanced by saturated fatty acids [69]. The gene expression of transmembrane protein 189 (TMEM189), which catalyzes the desaturation of PakEtn to PlsEtn, has been correlated with resistance to ferroptosis inducers in 654 cancer cell lines [69]. The administration of the plasmalogen precursor alkylglycerol decreased the growth, vascularization, and dissemination of Lewis lung carcinoma in a mouse model that used grafted tumors and reduced the plasmalogen content in the tumors [122].

## 4. Food Functions of Plasmalogens

### 4.1. Resources

Although the bovine brain is used as a resource for complex lipids, such as PlsEtn, its use has become difficult owing to outbreaks of bovine spongiform encephalopathy. Similarly, because eating the central nervous system of mammals poses anthropozoonotic risks, resources from different parts or different phyla are required.

Marine invertebrates, which exhibit lower neural differentiation than mammals, reportedly contain high levels of PlsEtn [6]. Accordingly, we investigated the contents of EtnGpl subclasses in marine invertebrates [7]. PlsEtn content in marine invertebrates varies widely (200–5000 nmol/g wet weight). The marine invertebrates examined, excluding prawns, squids, and octopuses, exhibited a high ratio of PlsEtn to EtnGpl in the muscle and viscera (65–98 mol%) (Figure 3). Sea anemones and starfish, which are not or are rarely used as food, had the highest content and ratio of PlsEtn (>2500 nmol/g wet weight, >95 mol%). The edible parts of ascidians, sea urchins, and shellfish have varying PlsEtn contents (200–4000 nmol/g wet weight) and a higher ratio of PlsEtn than mammalian brains (>65 mol%). PlsEtn from marine invertebrates is abundant in DHA and EPA at the *sn*-2 position; for example, the PlsEtn bearing DHA (18:0/22:6-PlsEtn) ratio of ascidian muscle and viscera is 9 mol% and 31 mol% of the total PlsEtn, respectively, whereas the ratio of land meat (leg muscle of beef, pig, and chicken) is <1 mol% (Figure 4) [8].

Chicken skin and large cervid meat (caribou and moose) have also been reported to have high levels of PlsEtn (1000 nmol and 5000 nmol/g wet weight, respectively; 75 mol% and 50 mol%, respectively), and the fatty acid composition of PlsEtn is predominantly ARA [123,124]. Bovine heart is often used as a PlsCho resource [125,126] and is also abundant in PakCho. Large cervid meat, beef, lamb, tuna, and scallops have been reported to contain high levels of PlsCho [124,127]. Some shellfish have high levels of serine-type plasmalogens, similar to those of PlsEtn [128]. Plasmalogens have not been found in plants and fungi, whereas those with unusual fatty chains and head groups have been found in anaerobic bacteria [129,130].

### 4.2. Alteration during Storage and Cooking

Alterations in the plasmalogen levels during storage and cooking have also been reported. GPLs in oysters are enzymatically hydrolyzed mainly to lyso-forms and free fatty acids during storage, depending on time and temperature, whereas PlsEtn hydrolysis is lower than that of other GPLs [131]. Similarly, plasmalogens in fish are lost during storage in a time- and temperature-dependent manner, and ethanolamine- and PUFA-type plasmalogen species can easily undergo degradation and oxidation [132]. During the postmortem aging (wet aging) of beef and venison, PlsEtn species are hardly hydrolyzed compared to PtdEtn bearing ARA [133]. Overall, GPLs in raw foods during storage and postmortem aging at low temperatures are altered by phospholipase hydrolysis (i.e., PLA_2_, PLC, and PLD) and oxidation. Plasmalogens hardly undergo enzymatic hydrolysis because of their substrate specificities, whereas PlsEtn with PUFAs is easily oxidized owing to its active methylene and amino groups compared to the species having saturated and monounsaturated fatty acids and PlsCho.

Among the cooking processes, boiling results in a lower loss of plasmalogens in beef compared with roasting and frying, which reduce plasmalogen levels depending on time and temperature [134]. The surface plasmalogen loss during frying was higher than the core loss, and the surface loss was reduced by butter coating. This indicates that, in roasting and frying processes, plasmalogens may be lost easily by dripping and oxidation. Additionally, it is thought that combinations with others affect plasmalogen conditions; for example, cooking and reducing sugar at the same time may modify EtnGpl, including PlsEtn, to Maillard reaction compounds (e.g., amadori-EtnGpl) [135].

### 4.3. Absorption and Metabolism

To utilize the dietary functions of plasmalogens effectively, their digestion, absorption, and metabolism must be understood. Because the vinyl ether bond at the *sn*-1 position of plasmalogen is acid-hydrolyzed, plasmalogen degradation by gastric acid is thought to occur; however, plasmalogens contained in diets are hardly degraded because of the buffering action of the diets [136,137]. In rats, ingested plasmalogens are absorbed into the lymph but not the portal vein [125]. PlsEtn is hydrolyzed to lyso-PlsEtn, which does not bear fatty acids at the *sn*-2 position, and is re-esterized after absorption; interestingly, PlsEtn, even though it hardly bears ARA like oyster- and ascidian-derived products, is preferentially re-esterized with ARA and partly base-converted to PlsCho during absorption, and structural changes occur in the intestinal mucosa [125,138,139]. Lyso-phosphatidylcholine acyltransferase 3, which plays a major role in incorporating ARA into phospholipids, is highly expressed in the intestine [140]. When PlsCho was administered, PlsCho levels in the lymph were four-fold higher than PlsEtn levels when PlsEtn was administered, and the absorbed PlsCho was slightly base-converted to PlsEtn [125]. After marine PlsEtn administration, the peak levels of PlsEtn containing PUFAs (i.e., DHA, EPA, and ARA) were almost the same in blood plasma and decreased early [138,139], suggesting that each level of PlsEtn with PUFAs in plasma is strictly controlled and rapidly transferred to the RBCs and organs [141]. By contrast, continuous diets containing bovine-brain-derived phospholipids (100 g of phospholipids/kg diets; 22 mol% plasmalogens in phospholipids) for 7 days increased the plasma and liver levels of plasmalogens but not RBCs [137].

### 4.4. Impact on Neurodegeneration

Some scientific studies have reported that the administration of plasmalogens improves cognitive deficits and lipid composition in rodent models and humans with neurodegenerative disorders, including AD. The administration of scallop-derived plasmalogens tends to improve cognitive function in patients with mild cognitive impairment (MCI), AD, and PD and increase PlsEtn levels in plasma and RBCs (Table 3) [11,142,143,144]. Ascidian-derived plasmalogens tended to improve memory function in subjects with MCI and mild forgetfulness [12,145]. Chicken-derived plasmalogens reportedly indicated similar tendencies in healthy subjects with mild forgetfulness [146,147]. As noted above, the administration of plasmalogens (0.5–100 mg/day) is expected to improve cognitive impairment. By contrast, the supplementation of DHA and EPA as *n*-3 PUFAs (0.2–2.0 g/day) is reported to attenuate the symptoms of MIC, mild AD, and PD [148]. Considering supplementation with other GPLs, 0.6–1.0 g/day of PtdCho and 0.3 g/day of phosphatidylserine are reported to improve the cognitive impairment associated with cerebrovascular disease and elderly memory impairment, respectively [149]. It is difficult to reference the advantages of marine plasmalogens for humans compared to those of other lipids because of the differences in dose, duration, evaluation, daily diet, and race of the subjects in the reports above.

In experiments using rodent models of AD, the oral administration of marine plasmalogens from ascidians, scallops, and sea cucumbers improves the cognitive function and biochemical characteristics of AD (Table 4). In aged mice, the oral administration of marine plasmalogens improves cognitive function and promotes neurogenesis [150]. Neuroinflammation induced by Aβ accumulation, lipopolysaccharides (LPSs), and a high-fat diet impairs peroxisome function, including DHAP-AT, and subsequently reduces PlsEtn levels [151,152]. A reduction in brain PlsEtn levels leads to suppression of neurogenesis by reducing the expression of the brain-derived neurotrophic factor (BDNF) [68] and the promotion of neuronal death by raising the expression of the p75 neurotrophin receptor (p75NTR) and protein kinase Cδ (PKCδ) [151,153]. p75NTR plays a critical role in the production of Aβ, neuronal death, neurite degeneration, and tau hyperphosphorylation; an increase in PKCδ expression is observed in human brains with AD, and the activation of PKCδ enhances toll-like receptor 4 (TLR4)-mediated pro-inflammatory signaling. In contrast, marine PlsEtn administration suppresses brain Aβ accumulation and tau hyperphosphorylation to attenuate neuroinflammation and apoptosis by restoring tropomyosin receptor kinase A/p75NTR signaling and attenuating PKCδ expression [151,153,154].

The oral and intraperitoneal administration of plasmalogens prepared from chicken breast muscle also attenuates inflammatory stress and enhances memory [68,152,155]. Although chicken-derived PlsEtn promotes BDNF expression by activating ERK and AKT signaling to induce neurogenesis in young mice, marine PlsEtn enhances memory function better [68]. This may be because marine PlsEtn is richer in DHA and EPA than the chicken derivative. Although EPA-rich PtdEtn also exhibits neuroprotective effects, these effects are weaker than those of marine PlsEtn [154].

The beneficial effects are thought to occur because the administration of plasmalogens maintains brain PlsEtn levels [68,155,156]. The levels of PlsEtn bearing DHA in the cerebral cortex correlate with working-memory-related learning ability in AD rats [156]. PlsEtn containing DHA exhibits stronger suppressive effects on neuronal inflammation, neuronal apoptosis, and Aβ aggregation in vitro than other PlsEtn species [97,157,158]. PlsEtn bearing DHA strongly reduces γ-secretase activity and reduces Aβ production in vitro [159]. Small-sized liposomes can deliver content to the brain through the blood–brain barrier [160], and intravenous injection of liposomes containing PlsEtn with DHA has been reported to increase its concentration in the prefrontal cortex and locomotor activity in normal rats [161].

The utilization of plasmalogen precursors and analogs has also been attempted to treat neurodegenerative disorders. These oral administrations improve the decreased content of dopamine-related substances in the brain and intestine and improve behavior in mouse models of PD and RCDP [162,163,164]. Their administration markedly increases PlsEtn levels in the serum or plasma but does not significantly increase brain levels. Because a slight but significant increase in brain PlsEtn levels was observed in an experiment using a labeled precursor [49], plasmalogens and/or precursors can pass through the blood–brain barrier, but in very small quantities [165]. In contrast to AD and aging, RCDP possesses plasmalogen synthesis defects, and PD does not markedly decrease brain PlsEtn levels [95,162]. Therefore, in PD and RCDP, plasmalogens may help patients recover from peripheral impairments and improve central function.

Hence, the ingestion of plasmalogens, particularly marine derivatives rich in DHA and EPA, is expected to attenuate the origin and development of cognitive impairments associated with aging. As described above, plasmalogen ingestion can rescue brain PlsEtn levels observed in AD models [155,156], and the mechanism suggests that the suppression of chronic inflammation in the entire body improves plasmalogen synthesis and degradation in the brain to increase plasmalogen levels rather than transfer from circulating plasmalogens.
molecules-28-06328-t004_Table 4Table 4Effects of administration of plasmalogens on age-associated cognitive impairments in animal studies.Resource and CompositionAnimalAdministrationEffectsRefs.Ascidian visceraEtnGpl composition: 80.4 mol% PlsEtnFatty composition: 22.0 mol% DHA, 37.1 mol% EPA, 7.6 mol% ARAFourth-generation male Wister rats that ate a fish-oil-deficient diet12 weeks oldAβ_40_ (4.9–5.5 nmol) and AlCl_3_ (0.5 μg) infusion into the cerebral ventricle and learning ability check for 12 weeksOral gavage of 200 mg (260 μmol) EtnGpl/kg/day for 6 wks from 24 weeks oldLong-term memory

; short-term memory

Plasma levels of 18:0/22:6-, 18:0:20:5-, and 18:0:20:4-PlsEtn

; RBC and liver levels of 18:0/22:6- and 18:0:20:5-PlsEtn

; cerebral cortex levels of 18:0/22:6-PlsEtn

[156]Egg yolkEtnGpl composition: 4.0 mol% PlsEtnFatty composition: 2.7 mol% DHA, 0.1 mol% EPA, 13.7 mol% ARALong-term memory⬌; short-term memory⬌Plasma, RBC, liver, and cerebral cortex levels of PlsEtn species⬌Sea cucumberPlsEtn composition: 93.4% PlsEtnFatty acid composition: 45.6% EPAMale SD rats6 weeks oldAβ_42_ (conc.: unclear) infusion into the cerebral ventricleOral gavage of 150 mg EtnGpl/kg/day for 26 daysMemory



In hippocampus: Aβ accumulation

; tau hyperphosphorylation



; inflammation



; apoptosis



In cortex: oxidative resistance



[154]Sea cucumber(PtdEtn enzymatically prepared from PtdCho)PtdEtn composition: 92.6% PtdEtnFatty acid composition: 49.3% EPA Memory

In hippocampus: Aβ accumulation

; tau hyperphosphorylation

; inflammation

; apoptosis

In cortex: oxidative resistance

ScallopPls composition: unclearFatty acid composition: 28.7% DHA, 26.1% EPA, 10.2% ARAMale C57/6J mice6 months oldi.p. LPS (250 μg/kg/day) for 7 days at 9 months oldDrinking water containing 0.1 μg/mL for 3 monthsIn cortex: PKCδ-positive microglial cells

[153]Male triple-transgenic mouse model of AD (PS1, tau, and APP)3 months oldDrinking water containing 1 μg/mL for 15 monthsIn cortex: PKCδ-positive microglial cells

; PKCδ protein

Sea cucumberPls composition: unclearMale APP/PS1 mice20 weeks oldA diet containing 0.1% PlsEtn for 16 weeksLong-term memory

; short-term memory

In hippocampus: Aβ generation

; soluble Aβ

; insoluble Aβ

; tau hyperphosphorylation

; neurodegeneration

; apoptosis

; lipid accumulation

; p75NTR

; TrkA phosphorylation

In cortex: Aβ accumulation

; apoptosis

In brain unclear part: oxidative resistance

In liver: lipid accumulation

; p75NTR

[151]Sea cucumberPlsEtn composition: 93.4% PlsEtnFatty acid composition: 11.4% DHA, 45.6% EPA, 10.1% ARAMale SAMP8 mice6 months oldA high-fat diet containing 1% PlsEtn for 2 monthsMemory

In hippocampus: Aβ generation

; soluble Aβ

; insoluble Aβ_40_

; insoluble Aβ_42_

In white matter: oxidative resistance

In brain unclear part: tau hyperphosphorylation

; glial activation

; inflammation

; apoptosis

; ARA content

[166]AscidianPls composition: unclearFemale C57BL/6J mice16 months oldOral gavage of 200 mg Pls/kg/day for 2 monthsMemory

In hippocampus: synaptic conditions (number, form, genesis)

; neurogenesis

; glial activation

; cytokine mRNA levels

[150]Chicken breast musclePls composition: 96.5% PlsEtn, 2.5% PlsChoFatty acid composition: 23.8% DHA, 0.9% EPA, 21.9% ARAMale C57BL/6 mice8 weeks oldDiet containing 0.01% Pls for 6 weeksMemory

In hippocampus: Pls level

; neurogenesis

[68]Chicken breast musclePls composition: 96.5% PlsEtn, 2.5% PlsChoFatty acid composition of PlsEtn: 23.8% DHA, 0.9% EPA, 21.9% ARADrinking water containing 0.01% Pls (*w*/*v*) for 6 weeksLong-term memory⬌; short-term memory

ScallopPls composition: 96.5% PlsEtn, 2.5% PlsChoFatty acid composition of PlsEtn: 37.1% DHA, 27.8% EPA, 24.9% ARALong-term memory

; short-term memory

Chicken breast musclePls composition: unclearMale C57/6J mice7 months oldi.p. LPS (250 μg/kg/day) for 7 days at 10 months oldDrinking water containing 0.1 or 10 μg Pls/mL for 3 monthsMemory

In cortex and hippocampus: Aβ accumulation

; glial activation

; cytokine mRNA levels


[152]Chicken breast musclePls composition: 47.6% PlsEtn (18.6% DHA, 24.9% ARA), 49.3% PlsCho (2.3% DHA, 17.2% ARA)Male C57/6J mice10 months oldi.p. LPS (250 μg/kg/day) for 7 daysi.p. 20 mg Pls/kg/day for 7 days along with LPS treatmentIn PFC and hippocampus: Aβ accumulation

; glial activation

; cytokine mRNA levels

; PlsEtn levels

[155]Aβ, amyloid-β; AD, Alzheimer’s disease; ARA, arachidonic acid; DHA, docosaxaenoic acid; EPA, eicosapentaenoic acid; EtnGpl, ethanolamine glycerophospholipid; LPS, lipopolysaccharide; PFC, prefrontal cortex; PKCδ, protein kinase Cδ; Pls, plasmalogen; PlsCho, choline plasmalogen; PlsEtn, ethanolamine plasmalogen; p75NTR, p75 neurotrophin receptor; TrkA, tropomyosin receptor kinase A; 

, decrease; 

 increase; ⬌, no change. 


### 4.5. Impact on Arteriosclerosis

Plasmalogen precursors and marine PlsEtn suppress the development of arteriosclerosis in rodent models. A plasmalogen precursor (butyl alcohol) suppressed the formation of atherosclerotic plaques in ApoE- or ApoE/glutathione peroxidase-1 (GPx1)-deficient mice fed a high-fat diet [72]. Additionally, precursor intake increases the plasma and cardiac levels of plasmalogens and lyso-phospholipids, which lose *sn*-1 fatty alcohols from plasmalogens by oxidation. In particular, in ApoE/GPx1-deficient mice, which possess lower resistance to oxidation, the precursor suppressed plasma levels of cholesterol and LDL and decreased oxidative stress and the expression of the vascular cellular adhesion molecule-1 in the aorta.

Sea-cucumber-derived PlsEtn intake attenuated the increase in plasma cholesterol and LDL levels to suppress the formation of atherosclerotic plaques in ApoE- or LDL-receptor-deficient mice fed a high-fat diet [167,168]. PlsEtn rich in EPA facilitated the synthesis and excretion of bile acids from excess cholesterol, whereas no effects were observed with the intake of EPA-rich PtdCho [167]. Likewise, in hamsters fed a high-fat diet, the administration of sea-urchin-derived PlsEtn and PakCho, which predominantly bear DHA and EPA, attenuated atherosclerotic lesions and hepatic steatosis by downregulating adipogenesis genes and upregulating lipid β-oxidation genes and bile acid biosynthesis genes in the liver; PlsEtn further increased hepatic sterol metabolism and serum levels of lipids bearing PUFAs compared to PakCho [169]. Additionally, dietary plasmalogens reduced hepatic cholesterol levels in rodents fed normal diets but not plasma cholesterol levels [137,170]. A decrease in the hepatic levels of PlsEtn and PlsCho bearing DHA and DHAP-AT expression was observed in a mouse model of nonalcoholic steatohepatitis (NASH), and cholesterol accumulation was induced in the livers of heterozygous DHAP-AT gene-deficient mice [171].

Overall, plasmalogens are responsible for the resistance to oxidation found in the vascular endothelium and hepatic sterol metabolism, and they act as antioxidants and LDL modulators in the vasculature to prevent arteriosclerosis. The structures of both the vinyl ether bond and EPA are important for suppressing arteriosclerosis.

### 4.6. Impact on Colon Impairments

The intestine digests food, absorbs nutrients and water, and is deeply implicated in the maintenance of human health via the immune and nervous systems [172]. However, intestinal impairments, such as colon cancer and inflammatory bowel disease (IBD), are becoming increasingly devastating diseases in all sexes worldwide, despite advances in diagnosis and treatment [173,174]. The incidence of IBD is increasing in younger generations and is thought to be caused by lifestyle factors, including diet, rather than aging. Aging and IBD increase the risk of developing colon cancer [175,176]. GPLs, particularly PtdCho, increase in colon cancer tumors by reducing their turnover [25]. Recently, PlsEtn metabolism was reported to be considerably dysregulated in dedifferentiated colon mucosa [177]. PlsCho has also been reported to inhibit cancer cell proliferation [178]. The authors indicated that plasmalogens might play a key role in protecting the normal colon mucosa from transitioning to hyper-proliferative and adenomatous polyps.

Marine plasmalogens suppress the formation of aberrant crypt foci (ACF) and intestinal inflammation in mouse models. A mouse model of ACF with precancerous colonic lesions was established via i.p. injection of 1,2-dimethylhydrazine (DMH). Diets containing ascidian-derived PlsEtn ameliorated DMH-induced ACF formation and oxidative and inflammatory stress in the colon [170]. A mouse model of IBD was established using drinking water containing sodium dextran sulfate (DSS). Dietary-ascidian-derived PlsEtn alleviated DSS-induced colon injury and neutrophil infiltration [179]. DMH and DSS treatments tended to decrease the PlsEtn species, particularly those bearing ARA, whereas ascidian-derived PlsEtn rescued their levels. In an in vitro intestinal tract model, ascidian-derived PlsEtn inhibited LPS-induced inflammation to protect intestinal cells [180]. Additionally, dietary PlsEtn improved cecal levels of short-chain fatty acids, which are metabolites of enteric bacteria and parameters of intestinal health and were decreased by DSS treatment [179]. Dietary PlsEtn rich in EPA altered enteric bacteria flora with a high-fat diet [168].

The in vivo and in vitro studies used fish oil and porcine-liver-derived PtdEtn for comparison. Marine PlsEtn performed better on intestinal inflammation caused by aging and other factors [170,179,180]. Therefore, the beneficial effects in the intestine are specific to PlsEtn bearing DHA and EPA.

## 5. Plasmalogens as Healthcare Materials

Among the materials used for the diagnosis, treatment, and management of human health and disease, plasmalogens have attractive functionalities as “healthcare materials”. In this section, the role of plasmalogens in the field of healthcare materials will be explained and examples of their use presented.

### 5.1. Materials for Functional Membrane of Biosensors

In recent years, “liquid biopsy” has gained attention as an important technique in the medical field for the early detection and diagnosis of biomarkers in the blood, including proteins, exosomes, small molecules, and circulating cells [181,182,183]. For the detection of biomarkers in liquid biopsy, low-cost and highly sensitive biosensors using semiconductor devices have been used. Such a biosensor adopts a functional membrane system that uses PtdCho. This system simulates the cell membrane and enables the evaluation of biomarkers in the blood and the assessment of compound permeability and adhesion to the cell membrane. Among these, the use of plasmalogens in functional membranes has garnered attention because it allows for the incorporation of detection sites such as antibodies, enzymes, and receptor molecules that cannot be incorporated with conventional PtdCho (Figure 5A) [184,185].

### 5.2. Materials for Light-Activated Liposomes

Bioactive compounds administered to the body via oral or intravenous administration achieve their activities after reaching the target organs or sites of inflammation. However, depending on the type of bioactive compound administered, its effectiveness may be limited because of instability or the body’s recognition of xenobiotic substances, raising concerns about the occurrence of unforeseen side effects [160]. One approach to solving such problems is to use a drug delivery system [186]. Drug delivery systems encapsulate bioactive components in formulated health materials such as liposomes and nanoparticles. This enhances therapeutic efficacy and reduces side effects. As introduced in Section 2 of this review, plasmalogens are characterized by the presence of an alkenyl (vinyl ether) linkage at the *sn-*1 position. Plasmalogens with this moiety have been reported to be characterized by their ability to alter membrane permeability in the process associated with the cleavage to lyso-phospholipids by photosensitized stimulation with laser light (Figure 5B) [187,188]. This mechanism allows the light-responsive release of bioactive compounds from plasmalogen-based liposomes. To date, the use of azobenzene to impart light sensitivity to liposomes has been limited in its clinical utility due to its high level of toxicity. On the other hand, plasmalogens exist in the body and have chemical structures that are more suitable and less toxic for healthcare use. Since ultraviolet and visible light reach only the superficial 1 cm layer of the skin but deeply into the eye, delivery studies targeting the retina are progressing using plasmalogen-based liposomes [189].

### 5.3. Materials for Nanoparticles with Endosomal Escape Capabilities

Depending on their size, surface structure, and charge, nanoparticles are taken up by cells via endocytosis and are subsequently degraded in the acidic environment of endosomes (pH < 6.5) [160,190]. The ability to escape from the endosome is essential for mRNA, drugs, bioactive substances, and other molecules to exert their effects in the cytoplasm. There are several ways to make nanoparticles achieve endosome escape, such as by using cationic amphiphiles, pH-sensitive polymers, and cell-penetrating peptides. During endocytosis, cationic amphiphiles attach and fuse with the anionic endosomal membrane, facilitating the release of bioactive compounds encapsulated in the nanoparticles into the cytoplasm (Figure 5C) [191]. Fay et al. reported that the transfection efficiency of mouse macrophage cell line (RAW 264.7) with plasmid DNA using polymeric nanoparticles modified with a cationic surfactant (dodecyltrimethylammonium bromide (DDAB)) was 1000 times higher compared to the commonly used lipofectamine [192]. Lipid nanoparticles can exhibit endosomal escape functionality by incorporating cationic lipids, such as plasmalogens [191,193]. Among the various available approaches, lipid nanoparticles are considered effective carriers for in vivo siRNA delivery [194]. Therefore, plasmalogens serve as valuable healthcare materials that provide endosomal escape functionality.

## 6. Conclusions

In this review, we discussed the beneficial effects of marine plasmalogens on aging-associated diseases, including cognitive impairment. Aging causes many metabolic impairments and increases oxidative and inflammatory stress, which decreases plasmalogen levels by suppressing biosynthesis and promoting degradation. A decrease in plasmalogen levels worsens aging-associated diseases via abnormal lipid signaling and reduced resistance to stress. The ingestion of marine plasmalogens, which are abundant in DHA and EPA, contributes to the maintenance or increase in plasmalogen levels in the body and intestines. Alterations in plasmalogen levels are due to exogenesis from ingested plasmalogens or endogenesis caused by reducing stress. Possessing the structures of both the vinyl ether bond and *n*-3 PUFAs is important for suppressing certain impairments. Plasmalogens possess chemical structures that are attractive for use as healthcare materials. We believe that the chronic administration of marine plasmalogens lowers the risk of age-associated diseases and has a unique capacity to profoundly improve quality of life.

## Figures and Tables

**Figure 1 molecules-28-06328-f001:**
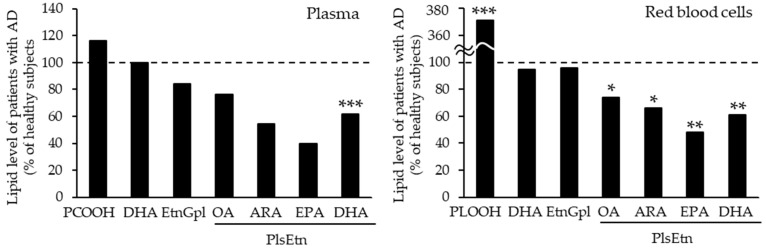
Alteration of lipid levels in blood of patients with AD [97]. Marks indicate significant differences between blood levels of patients with AD and their spouses (healthy subjects): *** *p* < 0.001, ** *p* < 0.01, * *p* < 0.05. AD, Alzheimer’s disease; ARA, arachidonic acid; DHA, docosahexaenoic acid; EPA, eicosapentaenoic acid; EtnGpl, ethanolamine glycerophospholipid; OA, oleic acid; PCOOH, phosphatidylcholine hydroperoxide; PLOOH, phospholipid hydroperoxide; PlsEtn, ethanolamine plasmalogen.

**Figure 2 molecules-28-06328-f002:**
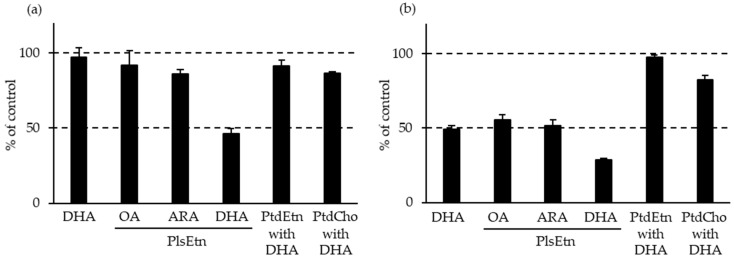
Effects of lipids on Aβ fibril formation [97]. (**a**) Aβ aggregation assay. (**b**) Aβ destabilization assay. Their values are expressed as a percentage of control aggregation, which was observed in the absence of 20 μM lipids. Aβ, amyloid β; ARA, arachidonic acid; DHA, docosahexaenoic acid; OA, oleic acid; PlsEtn, ethanolamine plasmalogen; PtdCho, phosphatidylcholine; PtdEtn, phosphatidylethanolamine.

**Figure 3 molecules-28-06328-f003:**
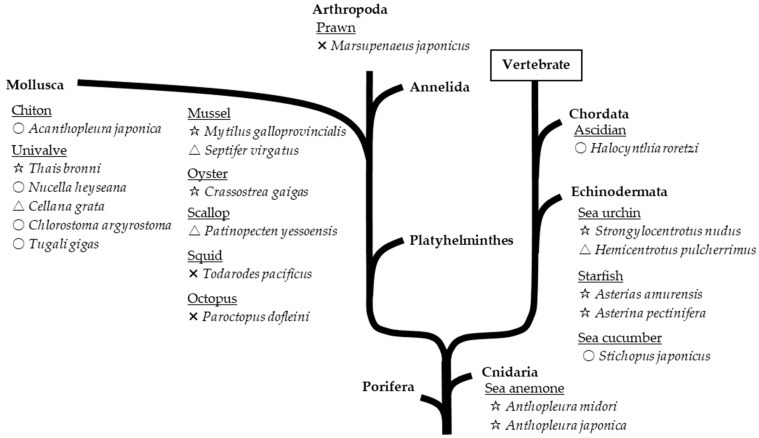
PlsEtn distribution in marine invertebrates on phylogenetic tree [7]. Marks indicate ratio of PlsEtn to ethanolamine glycerophospholipid: ☆ > 90 mol%, ○ 80–90 mol%, △ 50–80 mol%, × < 50 mol%. PlsEtn, ethanolamine plasmalogen.

**Figure 4 molecules-28-06328-f004:**
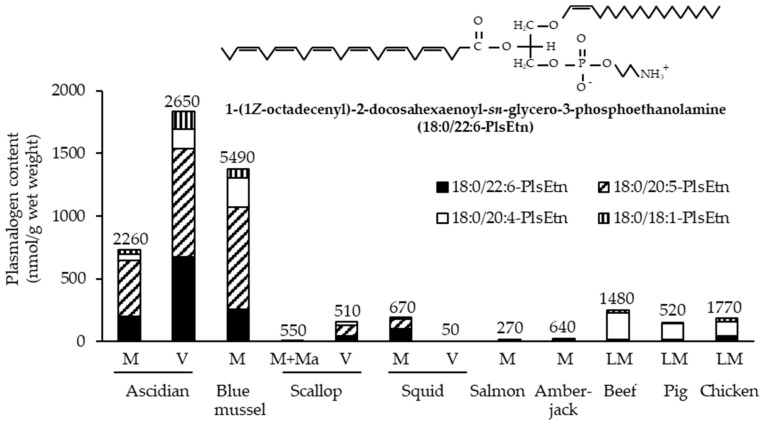
Composition of PlsEtn species in foodstuffs [8]. Numbers on bars indicate total PlsEtn content. M, muscle; Ma, mantle; LM, leg muscle; PlsEtn, ethanolamine plasmalogen; V, viscera.

**Figure 5 molecules-28-06328-f005:**
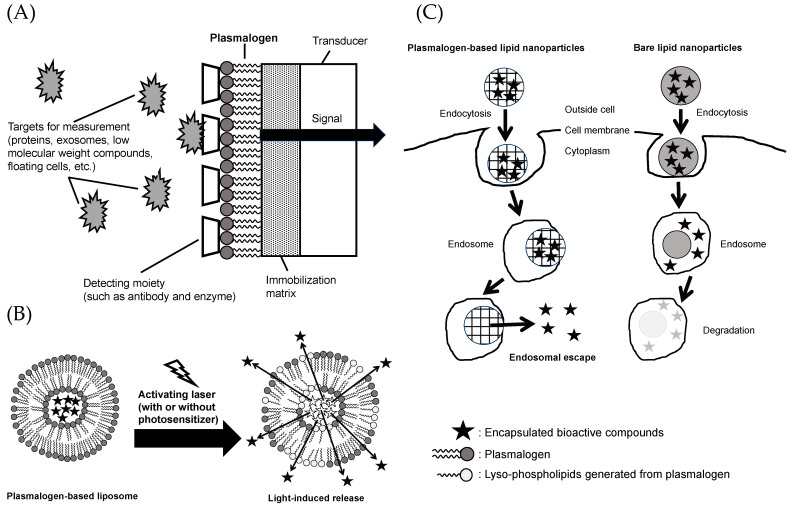
Application of plasmalogen as a healthcare material in (**A**) biosensors, (**B**) light-activated liposomes, and (**C**) nanoparticles with endosomal escape capabilities. (**A**) was modified with permission from [184] under the Creative Commons CC BY 4.0 license, https://creativecommons.org/licenses/by/4.0/ (accessed on 25 June 2023).

**Table 1 molecules-28-06328-t001:** Composition of plasmalogen in different mammalian organs and tissues.

Species	Organs and Tissues	PlsEtn (% EtnGpl)	PlsEtn (% Total PL)	PlsCho (% ChoGpl)	PlsCho (% Total PL)	Plasmalogen (% Total PL)	Plasmalogen (% Wet Weight)	Refs.
Human								
	Whole brain	58	20, 22	1	0.8, 0.9	22		[17,18]
	Brain prefrontal cortex	49		0.7				[19]
	Brain temporal cortex	46		0.5				
	Brain gray matter	49	19	0.1		19	0.4, 0.7	[20,21]
	Brain gray matter—frontal cortex	57				20	0.7	[10,22]
	Brain gray matter—parietal cortex	58				19	0.7	
	Brain gray matter—temporal cortex	56						
	Brain gray matter—cerebellum	63						
	Brain white matter	83, 86	33	2.6, 0.4	0.1	37	1.1, 1.8	[19,20,21]
	Brain white matter—frontal cortex	84				29	2	[10,22]
	Brain white matter—parietal cortex	81				36	2.7	
	Brain white matter—temporal cortex	83						
	Brain white matter—cerebellum	78						
	Myelin	91	36	0.5	0.1	43	7.0	[20]
	Heart	53, 51	17, 14, 15	26, 36	14, 11, 15	32, 29		[17,18,23]
	Skeletal muscle	48, 63	14, 15	19, 16	6.5, 9.7		0.2	[17,18,22,23]
	Cultured artery endothelial cells	48	13	1.9	0.9			[24]
	Cultured vein endothelial cells	34	8.7	0.2	0.1			
	Kidney	46	14	5	4.7			[17,18]
	Liver	8	4.7	3	3.4			
	Colon	36	11	9	3.9	19, 15	0.1, 0.1	[22,25]
	Lens	70	14					[26]
	Neutrophils	68		3.6				[27]
	Eosinophils	72		4				[28]
	Red blood cells	58, 48, 46	17,14, 9	8	2.8	20, 14, 9		[29,30,31]
	Plasma	53	2.5	5.5	5.2	7.7		[32]
	Serum		5.9		4.5	10		[33]
	HDL	55	6.4, 5	4.7	4.5, 4	11, 10		[32,34]
	LDL	60	5.9, 4	4.4	4.2, 4	10, 10		
	Platelet	54		0.8				[35]
	Cultured macrophage	62		8.8				[36]
	Spermatozoa		9		3	12		[37]
Rat								
	Whole brain	66		1.7			1	[19,22]
	Cortex		22					[38]
	Cerebellum		26					
	Hippocampus		23					
	Brainstem		32					
	Midbrain		24					
	Brain synaptic vesicles		16					[39]
	Heart	28	9.8	5.7	2.4	12	0.2	[22,40]
	Cultured heart sacrolemma	43		19				[41]
	Lung	42		1.6		16	0.3	[22,42]
	Kidney	20		2.3		12	0.3	
	Liver	3.3		0.4		3.4	0.07	
	Red blood cells	65	16	0.9	0.5	16		[13]
	Plasma	51, 36	0.4	0.5	0.4	0.8		[13,43]
	HDL	46		2.4				[43]
	LDL	32		1.9				
	VLDL	14		1.6				
	Mature spermatozoa	42		52		38		[44]
Mouse								
	Whole brain	47		1.2				[19]
	Cortex	46						[10,22]
	Cerebellum	53						
	Heart	24	8.2	62	31	39		[45]
Dog								
	Heart sarcolemma	73	16	57	37	53		[46]
Rabbit								
	Whole brain					26	1.3	[22]
	Heart					32	0.6	
	Lung					14	0.3	
	Kidney					14	0.3	
	Liver					2	0.06	
	Macrophage	61	13	6	2	15		[47]
Guinea pig								
	Whole brain						0.9	[21]

The composition of this table was primarily created by referring to [14,15] with their permission. For references [20,22], the values displayed in the table were converted from the data in the literature. Average molecular weight of glycerophospholipids used was 775. ChoGpl, choline glycerophospholipid; EtnGpl, ethanolamine glycerophospholipid; HDL, high-density lipoprotein; LDL, low-density lipoprotein; PL, phospholipid; PlsCho, choline plasmalogen; PlsEtn, ethanolamine plasmalogen; VLDL, very-low-density lipoprotein.

**Table 2 molecules-28-06328-t002:** Predominant plasmalogen species found in different mammalian organs and tissues.

Species	Organs and Tissues	Predominant Plasmalogen Species	Refs.
Human			
	Brain gray matter—frontal cortex	16:0/18:1-PlsEtn, 16:0/22:4-PlsEtn, 16:0/22:5-PlsEtn, 18:0/18:2-PlsEtn, 18:0/20:4-PlsEtn, 18:0/22:4-PlsEtn, 18:0/22:5-PlsEtn, 18:0/22:6-PlsEtn, 18:1/18:1-PlsEtn, 18:1/20:4-PlsEtn, 18:1/22:4-PlsEtn	[10]
	Brain gray matter—parietal cortex	16:0/18:1-PlsEtn, 16:0/22:4-PlsEtn, 16:0/22:5-PlsEtn, 18:0/18:2-PlsEtn, 18:0/20:4-PlsEtn, 18:0/22:4-PlsEtn, 18:0/22:5-PlsEtn, 18:0/22:6-PlsEtn, 18:1/18:1-PlsEtn, 18:1/20:4-PlsEtn, 18:1/22:4-PlsEtn	
	Brain gray matter—temporal cortex	16:0/22:4-PlsEtn, 16:0/22:5-PlsEtn, 16:0/22:6-PlsEtn, 18:0/18:2-PlsEtn, 18:0/20:4-PlsEtn, 18:0/22:4-PlsEtn, 18:0/22:5-PlsEtn, 18:0/22:6-PlsEtn, 18:1/18:1-PlsEtn, 18:1/20:4-PlsEtn, 18:1/22:4-PlsEtn	
	Brain gray matter—cerebellum	16:0/18:1-PlsEtn, 16:0/20:1-PlsEtn, 16:0/22:4-PlsEtn, 16:0/22:5-PlsEtn, 16:0/22:6-PlsEtn, 18:0/18:1-PlsEtn, 18:0/18:2-PlsEtn, 18:0/20:4-PlsEtn, 18:0/22:4-PlsEtn, 18:0/22:5-PlsEtn, 18:0/22:6-PlsEtn, 18:1/18:1-PlsEtn, 18:1/20:4-PlsEtn, 18:1/22:4-PlsEtn	
	Brain white matter—frontal cortex	16:0/18:1-PlsEtn, 16:0/20:1-PlsEtn, 16:0/22:4-PlsEtn, 16:0/22:5-PlsEtn, 18:0/18:1-PlsEtn, 18:0/20:4-PlsEtn, 18:0/22:4-PlsEtn, 18:0/22:5-PlsEtn, 18:1/18:1-PlsEtn, 18:1/20:1-PlsEtn, 18:1/20:4-PlsEtn, 18:1/22:4-PlsEtn	
	Brain white matter—parietal cortex	16:0/18:1-PlsEtn, 16:0/20:1-PlsEtn, 16:0/22:4-PlsEtn, 16:0/22:5-PlsEtn, 18:0/18:1-PlsEtn, 18:0/20:4-PlsEtn, 18:0/22:4-PlsEtn, 18:0/22:5-PlsEtn, 18:1/18:1-PlsEtn, 18:1/20:1-PlsEtn, 18:1/20:4-PlsEtn, 18:1/22:4-PlsEtn	
	Brain white matter—temporal cortex	16:0/18:1-PlsEtn, 16:0/20:1-PlsEtn, 16:0/22:4-PlsEtn, 16:0/22:5-PlsEtn, 18:0/18:1-PlsEtn, 18:0/20:4-PlsEtn, 18:0/22:4-PlsEtn, 18:0/22:5-PlsEtn, 18:1/18:1-PlsEtn, 18:1/20:1-PlsEtn, 18:1/20:4-PlsEtn, 18:1/22:4-PlsEtn	
	Brain white matter—cerebellum	16:0/18:1-PlsEtn, 16:0/20:1-PlsEtn, 16:0/22:4-PlsEtn, 16:0/22:5-PlsEtn, 18:0/18:1-PlsEtn, 18:0/20:4-PlsEtn, 18:0/22:4-PlsEtn, 18:0/22:5-PlsEtn, 18:1/18:1-PlsEtn, 18:1/20:1-PlsEtn, 18:1/20:4-PlsEtn, 18:1/22:4-PlsEtn	
	Cultured artery endothelial cells	16:0/20:4-PlsEtn, 16:0/22:4-PlsEtn, 16:0/22:5-PlsEtn, 16:0/22:6-PlsEtn, 18:0/20:4-PlsEtn, 18:1/20:4-PlsEtn	[24]
	Cultured vein endothelial cells	16:0/20:4-PlsEtn, 16:0/22:4-PlsEtn, 16:0/22:5-PlsEtn, 16:0/22:6-PlsEtn, 18:0/20:4-PlsEtn, 18:0/22:5-PlsEtn, 18:0/22:6-PlsEtn	
	Neutrophils	16:0/18:1-PlsEtn, 16:0/18:2-PlsEtn, 16:0/20:4-PlsEtn, 18:0/18:1-PlsEtn, 18:0/18:2-PlsEtn, 18:0/20:4-PlsEtn, 18:1/18:1-PlsEtn, 18:1/18:2-PlsEtn, 18:1/20:4-PlsEtn, 20:0/18:1-PlsEtn, 20:0/18:2-PlsEtn, 20:0/20:4-PlsEtn, 22:0/18:1-PlsEtn, 22:1/18:1-PlsEtn, 22:1/18:2-PlsEtn, 22:1/20:4-PlsEtn, 24:1/18:1-PlsEtn, 24:1/18:2-PlsEtn, 24:1/20:4-PlsEtn	[48]
	Lymphocytes	16:0/18:1-PlsEtn, 16:0/18:2-PlsEtn, 16:0/20:4-PlsEtn, 16:0/22:6-PlsEtn	[49]
	Serum	16:0/18:1-PlsEtn, 16:0/18:2-PlsEtn, 16:0/20:4-PlsEtn, 16:0/22:6-PlsEtn, 18:0/18:1-PlsEtn, 18:0/18:2-PlsEtn, 18:0/20:4-PlsEtn, 18:0/22:6-PlsEtn	[50]
	HDL	16:0/18:2-PlsEtn, 16:0/20:4-PlsEtn, 16:0/22:6-PlsEtn, 18:0/18:2-PlsEtn, 18:0/20:4-PlsEtn, 18:0/22:6-PlsEtn, 18:1/20:4-PlsEtn, 18:1/22:6-PlsEtn 16:0/18:1-PlsCho, 16:0/18:2-PlsCho, 16:0/20:4-PlsCho, 16:0/22:6-PlsCho, 18:0/18:2-PlsCho, 18:0/20:4-PlsCho, 18:1/20:4-PlsCho	[34]
	LDL	16:0/18:2-PlsEtn, 16:0/20:4-PlsEtn, 16:0/22:6-PlsEtn, 18:0/18:2-PlsEtn, 18:0/20:4-PlsEtn, 18:0/22:6-PlsEtn, 18:1/20:4-PlsEtn, 18:1/22:6-PlsEtn 16:0/18:1-PlsCho, 16:0/18:2-PlsCho, 16:0/20:4-PlsCho, 16:0/22:6-PlsCho, 18:0/18:2-PlsCho, 18:0/20:4-PlsCho	
	VLDL	16:0/18:2-PlsEtn, 16:0/20:4-PlsEtn, 16:0/22:6-PlsEtn, 18:0/20:4-PlsEtn, 18:0/22:6-PlsEtn, 18:1/20:4-PlsEtn, 18:1/22:6-PlsEtn 16:0/18:1-PlsCho, 16:0/18:2-PlsCho, 16:0/20:4-PlsCho, 16:0/22:6-PlsCho, 18:0/18:2-PlsCho, 18:0/20:4-PlsCho, 18:1/20:4-PlsCho	
	Platelet	16:0/18:2-PlsEtn, 16:0/20:4-PlsEtn, 18:0/20:4-PlsEtn 16:0/20:4-PlsCho, 18:0/20:4-PlsCho, 18:1/20:4-PlsCho	[35]
Rat			
	Cardiac sarcolemma	16:0/18:0-PlsEtn, 16:0/18:3-PlsEtn, 18:0/18:0-PlsEtn, 18:0/22:4-PlsEtn, 18:1/18:1-PlsEtn, 18:2/16:0-PlsEtn, 18:2/18:0-PlsEtn 16:0/18:0-PlsCho, 16:0/18:2-PlsCho, 16:0/18:3-PlsCho, 18:0/18:0-PlsCho, 18:0/20:4-PlsCho, 18:0/22:6-PlsCho, 18:1/22:6-PlsCho	[51]
Mouse			
	Cerebral cortex	16:0/18:1-PlsEtn, 16:0/20:1-PlsEtn, 16:0/22:6-PlsEtn, 18:0/18:1-PlsEtn, 18:0/20:4-PlsEtn, 18:0/22:6-PlsEtn, 18:1/16:0-PlsEtn, 18:1/18:1-PlsEtn, 18:1/20:4-PlsEtn, 18:1/22:6-PlsEtn	[52]
	Hippocampus	18:1/20:3-PlsEtn, 18:1/20:4-PlsEtn, 18:0/22:5-PlsEtn, 18:1/22:4-PlsEtn 16:0/18:1-PlsCho, 16:0/20:5-PlsCho, 16:0/22:5-PlsCho, 18:0/16:0-PlsCho	[53]
	Macrophage cell line RAW264.7	16:0/18:1-PlsEtn, 16:0/20:3-PlsEtn, 16:0/20:4-PlsEtn, 18:0/18:1-PlsEtn, 18:0/20:3-PlsEtn, 18:0/20:4-PlsEtn, 18:1/16:0-PlsEtn, 18:1/18:1-PlsEtn, 18:1/20:3-PlsEtn, 18:1/20:4-PlsEtn	[54]
Rabbit			
	Proximal tubules	16:0/18:1-PlsEtn, 16:0/20:4-PlsEtn, 18:0/18:2-PlsEtn, 18:0/20:4-PlsEtn, 18:1/18:2-PlsEtn, 18:1/20:4-PlsEtn	[55]
Dog			
	Heart sarcolemma	16:0/20:4-PlsEtn, 18:0/20:4-PlsEtn, 18:1/20:4-PlsEtn 16:0/18:1-PlsCho, 16:0/18:2-PlsCho, 16:0/20:4-PlsCho, 16:0/18:2-PlsCho, 18:1/18:1-PlsCho	[46]

Plasmalogen species greater than 5% of total PlsEtn or PlsCho are listed in the table. Only for the data from [48,49,50], all identified plasmalogen species were listed. The fatty chain combination is written as follows: 18:0/22:6-PlsEtn means 1-(1*Z*-octadecenyl)-2-docosahexaenoyl-*sn*-glycero-3-phosphoethanolamine. HDL, high-density lipoprotein; LDL, low-density lipoprotein; PlsCho, choline plasmalogen; PlsEtn, ethanolamine plasmalogen; VLDL, very-low-density lipoprotein.

**Table 3 molecules-28-06328-t003:** Effects of plasmalogen administration on age-associated diseases, including cognitive impairments, in clinical studies.

Resource and Dose	Subject	Effects	Refs.
Scallop Plasmalogen composition: unclear 1.0 mg/day for 24 weeks	Mild AD and MCI Placebo: *n* = 140, age 76.5, MMSE 24.2 1.0 mg: *n* = 145, age 76.4, MMSE 24.0	Not affected in whole In mild AD (MMSE 20–23) Female: WMS-R improvement compared with the placebo Younger than 78: WMS-R improvement compared with the placebo	[11]
Scallop Plasmalogen composition: unclear 1.0 mg/day for 24 weeks	MCI Placebo: *n* = 88, age 75.9, MMSE 25.6 1.0 mg: *n* = 90, age 75.8, MMSE 25.6	Improvement in total MMSE compared with before the intake Improvement in domain “orientation to place” compared with the placebo and before the intake Maintenance of domain “orientation to time” compared with before the intake	[143]
Scallop Plasmalogen composition: unclear 0.5 or 1.0 mg/day for 12 weeks	Moderate-to-severe AD 0.5 mg: *n* = 68, age 78.5, MMSE 13.0 1.0 mg: *n* = 74, age 76.6, MMSE 13.4	Improvement in total MMSE compared with before the intake Increase in plasma and RBC levels of PlsEtn compared with before the intake Increase in plasma PlsEtn levels in the 0.5 mg group compared with the 1.0 mg group Correlation between changes in MMSE and RBC PlsEtn	[144]
Scallop Ether lipid composition: 52% PlsEtn, 2% PlsCho, 4% PakEtn, 42% PakCho 1.0 mg/day for 24 weeks	PD PD: *n* = 10, age 67.8, MMSE 28.6	Improvement in total PDQ-39 compared with immediately before trial Deterioration of total PDQ-39 at 4 weeks later of the final administration Increase in plasma and RBC levels of PlsEtn compared with levels immediately before trial Decrease in RBC PlsEtn levels after 4 weeks without administration	[142]
Ascidian Plasmalogen composition: unclear 1.0 mg/day for 12 weeks	Healthy subjects with mild forgetfulness Placebo: *n* = 24, age 46.4, MMSE 29.0 1.0 mg: *n* = 25, age 45.6, MMSE 28.5	Improvement in Cognitrax domain “composite memory” at 8 and 12 weeks compared with the placebo	[145]
Ascidian Plasmalogen composition: 100% PlsEtn, trace PlsCho 0.5 or 1.0 mg/day for 12 weeks	MCI Placebo: *n* = 44, age 51.4, MMSE 25.3 0.5 mg: *n* = 45, age 51.5, MMSE 25.4 1.0 mg: *n* = 49, age 52.7, MMSE 25.7	Improvement in CogEvo domain “working memory performance” in the 1.0 mg group compared with the placebo In subjects older than 49 Improvement in total MMSE and “working memory performance” in the 1.0 mg group compared with the placebo	[12]
Chicken breast Plasmalogen composition: unclear 1.0, 10, or 100 mg/day for 12 weeks	Healthy subjects with mild forgetfulness Placebo: *n* = 17, age 58.7 1.0 mg: *n* = 15, age 57.2 10 mg: *n* = 16, age 59.9 100 mg: *n* = 15, age 59.0	Improvement in total RBANS in the 1.0 and 100 mg groups compared with before the intake Improvement in RBANS domain “attention” in all the groups, including the placebo, compared with before the intake Improvement in some Cognitrax domains in the plasmalogen groups compared with before the intake	[146]
Chicken breast Plasmalogen composition: unclear 0.5 or 1.0 mg/day for 12 weeks	Healthy subjects with mild forgetfulness age 50 to 79, MSSE > 26 Placebo: *n* = 24 0.5 mg: *n* = 25 1.0 mg: *n* = 25	Improvement in some CogEvo domains in all the groups, including the placebo, compared with before the intake In subjects older than 59 Improvement in some CogEvo domains in the 1.0 mg group compared with before the intake Improvement in CogEvo domain “verbal memory” in the 1.0 mg group compared with the placebo Improvement in CogEvo domain “psychomotor speed” in the 0.5 mg group compared with the placebo	[147]

AD, Alzheimer’s disease; MCI, mild cognitive impairment; MMSE, mini mental state examination; PakCho, 1-*O*-alkyl-2-acyl-*sn*-glycero-3-phosphocholine; PakEtn, 1-*O*-alkyl-2-acyl-*sn*-glycero-3-phosphoethanolamine; PD, Parkinson’s disease; PDQ-39, Parkinson’s disease questionnaire-39; PlsCho, choline plasmalogen; PlsEtn, ethanolamine plasmalogen; RBANS, repeatable battery for the assessment of neuropsychological status; RBC, red blood cell; WMS-R, Wechsler memory scale—revised.

## Data Availability

Not applicable.

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
