# Peer review of "Marine Plasmalogens: A Gift from the Sea with Benefits for Age-Associated Diseases"

_molecules, 2023, doi:10.3390/molecules28176328_

Round 1

Reviewer 1 Report

The review “Marine plasmalogens: The gift from sea with benefits for age-associated diseases” is a very interesting synthesis work dealing with a very important subject.

But to add interest to the review and extract more important information:

- The author needs to review this paragraph as it goes off-topic:

“Life expectancy and healthy life expectancy in the world have been increasing except 27 for the last several years, owing to unfortunate incidences, including the COVID-19 pan- 28 demic [1]. The latter is approximately 10 years shorter than the former, and it is hoped to 29 be close to the former. Moderate habits, including daily diet and exercise, are known to 30 prevent lifestyle diseases [2]. Also, in the post COVID-19 era, there is a prediction that the 31 importance of food bioactive compounds that maintain human body will increase with 32 the growing demand of the food industry [3].”

- The author needs to review this paragraph as it is difficult to understand and not clearly structured:

  “Ethanolamine and Choline GPLs (EtnGpl and ChoGpl), the major classes of GPLs 58 found in biological membranes of mammals, exist in three forms with alkyl, alkenyl, or 59 acyl linkages at the sn-1 position of the glycerol moiety: in case of EtnGpl, 1-O-alkyl-2- 60 acyl-sn-glycero-3-phosphoethanolamine (PakEtn), 1-O-alkenyl-2-acyl-sn-glycero-3-phos- 61 phoethanolamine (PlsEtn), and 1,2-diacyl-sn-glycero-3-phosphoethanolamine (PtdEtn), 62 respectively: in case of ChoGpl, PakCho, PlsCho, and PtdCho, respectively. The 63 alkenylacyl form is called a plasmalogen. Fatty alcohols at the sn-1 position of plasmalo- 64 gens consist primarily of C16:0 (palmitoyl), C18:0 (stearoyl), or C18:1 (oleoyl) carbon 65 chains, whereas those at the sn-2 position consist primarily of PUFAs, such as DHA and 66 arachidonic acid (ARA). 67”

- In the Biological Role section, I propose to add the biological role of each organ rich in Plasmalogens:

Biological role in brain 

Biological role in heart

Biological role in blood

- The author has to review this sentence because it is like a description for a histogram! 

  The author has to make a synthesis of the studies and bring out a well-founded scientific idea (clear and understandable)

  In PD brain, plasmalogens were 182 lower than those in the control but not significantly; however, in lipid rafts from the PD 183 brain, the levels of plasmalogens were significantly lower [51, 57]. Additionally, PlsEtn 184 levels in the plasma and RBCs of patients with PD were significantly lower than those in 185 control subjects [58, 59]. 186

- Regarding the biological role, I suggest explaining the role and connection between plasmalogen and disease, rather than just mentioning the degree of variation.

- The author needs to develop this paragraph (and add a title for this section) with more scientific explanation 

« Alterations in the plasmalogen levels during storage and cooking have also been re- 253 ported. GPLs in oysters are enzymatically hydrolyzed mainly to the lyso-forms and free 254 fatty acids during storage, depending on time and temperature, whereas PlsEtn hydroly- 255 sis is lower than that of other GPLs [88]. Similarly, plasmalogens in fish are lost during 256 storage in a time- and temperature-dependent manner, and ethanolamine- and PUFA- 257 type plasmalogen species easily undergo degradation and oxidation [89]. During the post- 258 mortem aging of beef and venison, PlsEtn species are hardly hydrolyzed compared to 259 PtdEtn-bearing ARA [90]. Among the cooking processes, boiling results in a lower loss of 260 plasmalogens in beef compared with roasting and frying, which reduce plasmalogen lev- 261 els depending on time and temperature [91] ».

- In this review, the authors want to discuss Marine plasmalogens but the tables and studies used, focused more on other sources of plasmalogens 

I think the author needs to focus more on his subject.

- The review isn’t well-structured it needs a logical succession of information

- The author needs to add molecular regulation for plasmalogen use .

The author has to show the major revisions in the text, with a different color text, by highlighting the changes.

Author Response

Response to Reviewer#1

Comments: The review “Marine plasmalogens: The gift from sea with benefits for age-associated diseases” is a very interesting synthesis work dealing with a very important subject.

But to add interest to the review and extract more important information. The author has to show the major revisions in the text, with a different color text, by highlighting the changes.

Response: We express our appreciation to Reviewer#1 for the positive comments, which have helped us to improve our manuscript. We hope that the revised manuscript will be suitable for publication.

Comment 1: - The author needs to review this paragraph as it goes off-topic:

“Life expectancy and healthy life expectancy in the world have been increasing except for the last several years, owing to unfortunate incidences, including the COVID-19 pandemic [1]. The latter is approximately 10 years shorter than the former, and it is hoped to be close to the former. Moderate habits, including daily diet and exercise, are known to prevent lifestyle diseases [2]. Also, in the post COVID-19 era, there is a prediction that the importance of food bioactive compounds that maintain human body will increase with the growing demand of the food industry [3].”

Response: We deleted this sentence and slightly revised the following sentence.

P1 L28-32

“Aging decreases metabolic, clearance, and immune functions, and increases oxi-dative and inflammatory stress [1, 2]. For example, a decrease in clearance of amy-loid-β (Aβ) from brain by aging is hypothesized to develop into Alzheimer’s disease (AD) via chronic oxidation and inflammation [3]. As human life expectancy is high in contemporary society, it is necessary to reduce the risk of age-associated diseases.”

Comment 2: - The author needs to review this paragraph as it is difficult to understand and not clearly structured:

“Ethanolamine and Choline GPLs (EtnGpl and ChoGpl), the major classes of GPLs found in biological membranes of mammals, exist in three forms with alkyl, alkenyl, or acyl linkages at the sn-1 position of the glycerol moiety: in case of EtnGpl, 1-O-alkyl-2-acyl-sn-glycero-3-phosphoethanolamine (PakEtn), 1-O-alkenyl-2-acyl-sn-glycero-3-phos-phoethanolamine (PlsEtn), and 1,2-diacyl-sn-glycero-3-phosphoethanolamine (PtdEtn), respectively: in case of ChoGpl, PakCho, PlsCho, and PtdCho, respectively. The alkenylacyl form is called a plasmalogen. Fatty alcohols at the sn-1 position of plasmalogens consist primarily of C16:0 (palmitoyl), C18:0 (stearoyl), or C18:1 (oleoyl) carbon chains, whereas those at the sn-2 position consist primarily of PUFAs, such as DHA and arachidonic acid (ARA).”

Response: To clear this sentence, we revised as follows,

P2 L52-65

“GPLs are composed of a glycerol bone with two fatty chains and a polar head with phosphate. They are classified by their polar head. For example, ethanolamine and choline GPLs (EtnGpl and ChoGpl), the major classes of GPLs found in the biological membranes of mammals, possess phosphoethanolamine and phosphocholine as the polar head, respectively. Moreover, GPLs are classified in three subclasses with alkyl, alkenyl, or acyl linkages at the sn-1 position of the glycerol moiety. In case of EtnGpl, the subclasses are 1-O-alkyl-2-acyl-sn-glycero-3-phosphoethanolamine (PakEtn), 1-O-alkenyl-2-acyl-sn-glycero-3-phosphoethanolamine (PlsEtn), and 1,2-diacyl-sn-glycero-3-phosphoethanolamine (PtdEtn), respectively. In case of ChoGpl, the subclasses are 1-O-alkyl-2-acyl-sn-glycero-3-phosphocholine (PakCho), 1-O-alkenyl-2-acyl-sn-glycero-3-phosphocholine (PlsCho), and phosphatidylcholine (PtdCho), respectively. The alkenylacyl form is called a plasmalogen. Fatty alcohols at the sn-1 position of plasmalogens consist primarily of C16:0 (palmitoyl), C18:0 (stearoyl), or C18:1 (oleoyl) carbon chain, whereas those at the sn-2 position consist primarily of PUFAs such as DHA and arachidonic acid (ARA).”

Comment 3: - In the Biological Role section, I propose to add the biological role of each organ rich in Plasmalogens:

Biological role in brain

Biological role in heart

Biological role in blood

Response: The Biological Role was recomposed.

P6 L115-P7 L141

“The significance of plasmalogens in the brain and nerves has been extensively in-vestigated. Because PlsEtn easily forms an inverse-hexagonal phase at body tempera-ture compared with PtdEtn, PlsEtn is involved in membrane fusion during synaptic transmission [66]. In plasmalogen-deficient mice (knockout of the DHAP-AT gene), myelination in the spinal cord and optic nerves was improved by intraperitoneal (i.p.) injection of alkyl glycerol [67]. The knockdown of DHAP-AT gene in the hippocampus of adult mice decreases memory function and neurogenesis [68]. According to a recent study, a decrease in PlsEtn levels reduces cholesterol synthesis; because the brain does not receive cholesterol from the blood, low PlsEtn levels in the brain lead to a shortage of cholesterol and metabolite 24-hydroxychlesterol as a ligand of the liver X receptor (LXR), which inhibit myelin formation via LXR [64]. Additionally, plasmalogens are reported to be related to ferroptosis, a type of cell death caused by the accumulation of phospholipid hydroperoxides. Saturated fatty acids, which are metabolized to fatty alcohols by Far1, enhance the effects of ferroptosis-inducers [69]. Because Far1 levels receive feedback inhibition by plasmalogens, such as those described above, a decrease in the brain levels of plasmalogens may induce neurocyte death.

In terms of the action of plasmalogens on the heart and blood, there are reports on its antioxidant properties [70-72]. In macrophages, which are derived from monocytes and are responsible for the immune response, PlsEtn affects the signal transduction of inflammation by providing ARA as a source of eicosanoids [73] and by controlling the number and size of lipid rafts [51]. Because plasmalogens in heart and blood plasma possess the same levels of PlsCho as PlsEtn, whereas those in other organs are pre-dominantly PlsEtn, PlsCho may play important roles in the heart and blood [74]. Mice deficient in mitochondrial transacylase tafazzin, a model of Barth syndrome, de-creased cardiac levels of PlsCho, not PlsEtn, whereas a significant decrease in PlsEtn levels was observed in the brain, kidney, and liver [45, 75]. However, the biological significance of plasmalogens in the organs remains unclear.”

Comment 4: - The author has to review this sentence because it is like a description for a histogram!

The author has to make a synthesis of the studies and bring out a well-founded scientific idea (clear and understandable)

In PD brain, plasmalogens were lower than those in the control but not significantly; however, in lipid rafts from the PD brain, the levels of plasmalogens were significantly lower [51, 57]. Additionally, PlsEtn levels in the plasma and RBCs of patients with PD were significantly lower than those in control subjects [58, 59].

Response: The sentence was added.

P9 L205-215

“In the PD brain, plasmalogens were lower than those in the control brain, but not significantly; however, in lipid rafts from the PD brain, the levels of plasmalogens were significantly lower [95, 100]. Lipid rafts are strongly associated with signal transduction. It has been suggested that a reduction in plasmalogen levels decreases membrane strength and levels of eicosanoids and docosanoids from PUFAs, which affect signal transduction. Additionally, PlsEtn levels in the plasma and RBCs of patients with PD are significantly lower than those in control subjects [101, 102]. The lipid composition of blood is thought to reflect that of the organs [103]. In contrast, reduction of blood PlsEtn rich in PUFAs may induce impairment of brain blood flow, which associated with PD [104]. It would be interesting to determine whether, in patients with PD, blood PlsEtn levels fall earlier than those of brain lipid rafts.”

Comment 5: - Regarding the biological role, I suggest explaining the role and connection between plasmalogen and disease, rather than just mentioning the degree of variation.

I think the author needs to focus more on his subject.

- The review isn’t well-structured it needs a logical succession of information

- The author needs to add molecular regulation for plasmalogen use.

Response: Their sentences were revised.

P14 L365-389

“ In experiments using rodent models of AD, oral administration of marine plasmalogens from ascidians, scallops, and sea cucumbers improved cognitive function and biochemical characteristics of AD (Table 4). In aged mice, oral administration of marine plasmalogens improves cognitive function and promotes neurogenesis [150]. Neuroinflammation induced by Aβ accumulation, lipopolysaccharide (LPS), and a high-fat diet impairs peroxisome function including DHAP-AT and subsequently re-duces PlsEtn levels [151, 152]. Reduction in the brain PlsEtn levels leads to suppression of neurogenesis by reducing expression of brain-derived neurotrophic factor (BDNF) [68] and promotion of neuronal death by raising the expression of p75 neurotrophin receptor (p75NTR) and protein kinase Cδ (PKCδ) [151, 153]; p75NTR plays a critical role in production of Aβ, neuronal death, neurite degeneration, and tau hyperphos-phorylation; an increase in PKCδ expression is observed in human brains with AD, and activation of PKCδ enhances toll-like receptor 4 (TLR4)-mediated pro-inflammatory signaling. In contrast, marine PlsEtn administration suppressed brain Aβ accumulation and tau hyperphosphorylation to attenuate neuroinflammation and apoptosis by restoring tropomyosin-receptor kinase A/p75NTR signaling and attenuating PKCδ ex-pression [151, 153, 154].

Oral and intraperitoneal administration of plasmalogens prepared from chicken breast muscle also attenuates inflammatory stress and enhances memory [68, 152, 155]. Although chicken-derived PlsEtn promotes BDNF expression by activating ERK and AKT signaling to induce neurogenesis in young mice, marine PlsEtn enhances memory function better [68]. This may be because marine PlsEtn is richer in DHA and EPA than the chicken derivative. Although EPA-rich PtdEtn also exhibits neuroprotective effects, these effects are weaker than those of marine PlsEtn [154].

The beneficial effects are thought to occur because the administration of plasmalogens maintains the brain PlsEtn levels [68, 155, 156].”

Comment 6: - The author needs to develop this paragraph (and add a title for this section) with more scientific explanation

« Alterations in the plasmalogen levels during storage and cooking have also been reported. GPLs in oysters are enzymatically hydrolyzed mainly to the lyso-forms and free fatty acids during storage, depending on time and temperature, whereas PlsEtn hydrolysis is lower than that of other GPLs [88]. Similarly, plasmalogens in fish are lost during storage in a time- and temperature-dependent manner, and ethanolamine- and PUFA-type plasmalogen species easily undergo degradation and oxidation [89]. During the postmortem aging of beef and venison, PlsEtn species are hardly hydrolyzed compared to PtdEtn-bearing ARA [90]. Among the cooking processes, boiling results in a lower loss of plasmalogens in beef compared with roasting and frying, which reduce plasmalogen levels depending on time and temperature [91] ».

Response: The new section “4.2 Alteration during Storage and Cooking” was made and scientific sentences were added.

P11 L298-P12 L319

“4.2 Alteration during Storage and Cooking

Alterations in the plasmalogen levels during storage and cooking have also been reported. GPLs in oysters are enzymatically hydrolyzed mainly to lyso-forms and free fatty acids during storage, depending on time and temperature, whereas PlsEtn hy-drolysis is lower than that of other GPLs [131]. Similarly, plasmalogens in fish are lost during storage in a time- and temperature-dependent manner, and ethanolamine- and PUFA-type plasmalogen species can easily undergo degradation and oxidation [132]. During the postmortem aging (wet-aging) of beef and venison, PlsEtn species are hardly hydrolyzed compared to PtdEtn bearing ARA [133]. Overall, GPLs in raw foods during storage and postmortem aging at low temperatures are altered by phospho-lipase hydrolysis (i.e., PLA2, PLC, and PLD) and oxidation. Plasmalogens hardly un-dergo enzymatic hydrolysis because their substrate specificities, whereas PlsEtn with PUFAs is easily oxidized owing to its active methylene and amino group compared to the species having saturated and monounsaturated fatty acids and PlsCho.

Among the cooking processes, boiling results in a lower loss of plasmalogens in beef compared with roasting and frying, which reduce plasmalogen levels depending on time and temperature [134]. The surface plasmalogen loss during frying was higher than the core loss, and the surface loss was reduced by butter coating. This indicates that in roasting and frying processes, plasmalogens may be lost easily by dripping and oxidation. Additionally, it is thought that combination with others affects plasmalogen conditions; for example, cooking with reducing sugar may modify EtnGpl including PlsEtn to Maillard-reaction compounds (e.g., amadori-EtnGpl) [135].”

Comment 7: - In this review, the authors want to discuss Marine plasmalogens but the tables and studies used, focused more on other sources of plasmalogens

Response: We discuss marine plasmalogens compared to other sources.

P12 L349-358

“As noted above, administration of plasmalogens (0.5–100 mg/day) is expected to im-prove cognitive impairment. By contrast, supplementation of DHA and EPA as n-3 PUFAs (0.2–2.0 g/day) is reported to attenuate the symptoms of MIC, mild AD, and PD [148]. Considering supplementation with other GPLs, 0.6–1.0 g/day of PtdCho and 0.3 g/day of phosphatidylserine are reported to improve cognitive impairment associated with cerebrovascular disease and elderly memory impairment, respectively [149]. It is difficult to reference the advantages of marine plasmalogens for humans compared to those of other lipids because of the differences in dose, duration, evaluation, daily diet, and race of subjects among the above reports.”

P14 L381-387

“Oral and intraperitoneal administration of plasmalogens prepared from chicken breast muscle also attenuates inflammatory stress and enhances memory [68, 152, 155]. Although chicken-derived PlsEtn promotes BDNF expression by activating ERK and AKT signaling to induce neurogenesis in young mice, marine PlsEtn enhances memory function better [68]. This may be because marine PlsEtn is richer in DHA and EPA than the chicken derivative. Although EPA-rich PtdEtn also exhibits neuroprotective effects, these effects are weaker than those of marine PlsEtn [154].”

Reviewer 2 Report

This review paper provides a comprehensive examination of plasmalogens, covering their biological significance, relationship with age-associated diseases, food functions, and potential as healthcare materials. It offers an overview of their structures, distribution, and role as endogenous antioxidants in the nervous system. The correlation between plasmalogen levels and age-associated diseases, particularly Alzheimer's and Parkinson's diseases, arteriosclerosis and cancer, is explored, along with the potential benefits of increasing plasmalogen intake to prevent or treat such conditions.

Additionally, the paper discusses the food sources of plasmalogens, including marine invertebrates, and their potential applications as healthcare materials. The overall aim of the manuscript is to present a comprehensive understanding of plasmalogens and their potential health advantages.

Overall, the paper is sufficiently unbiased, providing a well-rounded view of current research on plasmalogens.

Though the paper highlights the biological significance, relationships with age-associated diseases, food functions, and healthcare potential of plasmalogens, it does not directly compare marine plasmalogens to other sources of glycerophospholipids. Nonetheless, it does discuss the contribution of marine plasmalogens to maintaining health during aging.

1) To improve readability for general readers, adding a table that reports the levels of plasmalogen in different mammalian organs or structures would be beneficial.

2) Could the Authors provide information about the predominant plasmalogen species found in various human and/or animal organs?

3) In chapter 3, which discusses "Alterations of Plasmalogens in Aging and Associated Diseases), it is mentioned that changes in plasmalogen metabolism may contribute to the development of various types of cancer. However, in the review, only colon cancer is mentioned.

It would be helpful to review the associations of plasmalogens in gastrointestinal cancers as well as other types of cancers.

4) It would be worth discussing whether variations in the quality and quantity of plasmalogens might be associated with other chronic inflammatory conditions. This suggestion raises from the consideration that cancer, whose correlation with plasmalogens levels is discussed in chapter 3, cannot be exclusively considered an age-related disease.

5) Regarding the manuscript's structure, in chapter 4, which delves into "Food Functions of Plasmalogens" there is a paragraph entitled "Impact on Colon Impairments" that lacks adequate "introduction" in other sections of the paper.

It would be beneficial to introduce this paragraph by referring (in chapter 3?) to potential roles of plasmalogens in gut physiology and pathological conditions.

6) In the Conclusions section, the Authors might express their opinion on whether the observed drop in plasmalogen levels during aging and in disease is a cause or an effect, providing further insight into their findings.

Author Response

Response to Reviewer#2

Comments: This review paper provides a comprehensive examination of plasmalogens, covering their biological significance, relationship with age-associated diseases, food functions, and potential as healthcare materials. It offers an overview of their structures, distribution, and role as endogenous antioxidants in the nervous system. The correlation between plasmalogen levels and age-associated diseases, particularly Alzheimer's and Parkinson's diseases, arteriosclerosis and cancer, is explored, along with the potential benefits of increasing plasmalogen intake to prevent or treat such conditions.

Additionally, the paper discusses the food sources of plasmalogens, including marine invertebrates, and their potential applications as healthcare materials. The overall aim of the manuscript is to present a comprehensive understanding of plasmalogens and their potential health advantages.

Overall, the paper is sufficiently unbiased, providing a well-rounded view of current research on plasmalogens.

Though the paper highlights the biological significance, relationships with age-associated diseases, food functions, and healthcare potential of plasmalogens, it does not directly compare marine plasmalogens to other sources of glycerophospholipids. Nonetheless, it does discuss the contribution of marine plasmalogens to maintaining health during aging.

Response: We thank the reviewer for helpful suggestions, especially for supplying us with better terms and improved sentences. We have revised the manuscript based on the Reviewer's comments.

Comment 1: To improve readability for general readers, adding a table that reports the levels of plasmalogen in different mammalian organs or structures would be beneficial.

Could the Authors provide information about the predominant plasmalogen species found in various human and/or animal organs?

Response: We appreciate and agree with your valuable comments. We added the following two TABLES:

・TABLE1: Composition of plasmalogen in different mammalian organs and tissues.

・TABLE2: Predominant plasmalogen species found in different mammalian organs and tissues.

Comment 2: In chapter 3, which discusses "Alterations of Plasmalogens in Aging and Associated Diseases), it is mentioned that changes in plasmalogen metabolism may contribute to the development of various types of cancer. However, in the review, only colon cancer is mentioned.

It would be helpful to review the associations of plasmalogens in gastrointestinal cancers as well as other types of cancers.

Response: Other cancer information was added.

P9 L242-P10 L262

“Many studies have reported that higher levels of ether lipids (i.e., plasmalogens and alkyl-GPLs) are observed in cancer cells and tumors than in normal cells and tissues. This is due to the upregulation of biosynthesis enzymes of ether lipids in malignant cells, and ether lipids contributing to promoting cancer progression via the generation of signaling lipids [114]. Hypoxia, which is a factor of cancer progression, increases plasmalogen levels in in vitro colon cancer cells by activating re-acylation of lyso-plasmalogens, not plasmalogen synthesis [115]. Patients with gastric carcinoma show higher plasma levels of PlsEtn [116]. In contrast, some studies reported that cancers decreased plasmalogen levels, esophageal tumors decreased PlsEtn and other phospholipids [117], and patients with pancreatic cancer exhibited lower serum levels of PlsEtn than control subjects [118]. Lyso-PlsEtn and n-3 PUFAs, provided from PlsEtn by epidermal secretory PLA2 (sPLA2-IID), promote skin cancer hypergrowth and re-duce anti-tumor immunity, respectively [119]. Additionally, ferroptosis, an attractive target for anticancer therapy [120], is associated with plasmalogens. Plasmalogens containing PUFAs accelerate ferroptosis via lipid peroxidation [121], while plasmalogens with non-PUFAs suppress ferroptosis enhanced by saturated fatty acids [69]. Gene expression of transmembrane protein 189 (TMEM189), which catalyzes the de-saturation of PakEtn to PlsEtn, has been correlated with resistance to ferroptosis in-ducers in 654 cancer cell lines [69]. Administration of the plasmalogen precursor alkylglycerol decreased the growth, vascularization, and dissemination of Lewis lung carcinoma in a mouse model that used grafted tumors, and reduced the plasmalogen content in the tumors [122].”

Comment 3: It would be worth discussing whether variations in the quality and quantity of plasmalogens might be associated with other chronic inflammatory conditions. This suggestion raises from the consideration that cancer, whose correlation with plasmalogens levels is discussed in chapter 3, cannot be exclusively considered an age-related disease.

Regarding the manuscript's structure, in chapter 4, which delves into "Food Functions of Plasmalogens" there is a paragraph entitled "Impact on Colon Impairments" that lacks adequate "introduction" in other sections of the paper.

It would be beneficial to introduce this paragraph by referring (in chapter 3?) to potential roles of plasmalogens in gut physiology and pathological conditions.

Response: The introduction and information on inflammation without aging were added.

P7 L131-134

“In macrophages, which are derived from monocytes and are responsible for the immune response, PlsEtn affects the signal transduction of inflammation by providing ARA as a source of eicosanoids [73] and by controlling the number and size of lipid rafts [51].”

P17 L453-458

“The intestine digests food, absorbs nutrients and water, and is deeply implicated in the maintenance of human health via the immune and nervous systems [172]. However, intestinal impairments, such as colon cancer and inflammatory bowel dis-ease (IBD), are becoming increasingly devastating diseases in all sexes worldwide, de-spite advances in diagnosis and treatment [173, 174]. The incidence of IBD is increasing in younger generations and is thought to be caused by life-style factors, including diet, rather than aging. Aging and IBD increase the risk of developing colon cancer [175, 176].”

P18 L478-479

“Marine PlsEtn performed better on intestinal inflammation caused by aging and other factors [170, 179, 180].”

Comment 4: In the Conclusions section, the Authors might express their opinion on whether the observed drop in plasmalogen levels during aging and in disease is a cause or an effect, providing further insight into their findings.

Response: The conclusions section was revised.

P19 L542-P20 L554

“In this review, we discussed the beneficial effects of marine plasmalogens on ag-ing-associated diseases, including cognitive impairment. Aging causes many metabolic impairments and increases oxidative and inflammatory stress, which occurs decreases plasmalogen levels by suppressing biosynthesis and promoting degradation. A de-crease in plasmalogen levels worsens aging-associated diseases via abnormal lipid signaling and reduced resistance to stress. The ingestion of marine plasmalogens, which are abundant in DHA and EPA, contributes to the maintenance or increase of plasmalogen levels in the body and intestines. Alterations in plasmalogen levels are due to exogenesis from ingested plasmalogens or endogenesis caused by reducing stress. Possessing the structures of both the vinyl ether bond and n-3 PUFA is important for suppressing certain impairments. Plasmalogens possess chemical structures that are attractive for use as healthcare materials. We believe that chronic ad-ministration of marine plasmalogens lowers the risk of age-associated diseases and has a unique capacity to profoundly improve quality of life.”

Round 2

Reviewer 1 Report

In this version of the review, “Marine plasmalogens: The gift from sea with benefits for age associated diseases.” We can see an acceptable evolution compared to the first version because it has become more structured with more explanation.

the authors have relatively taken the reviewer's remarks and suggestions into consideration, which has positively impacted the quality and consistency of the article.

with this version, the article shows an excellent scientific level and represents an added value in the interested research topics 

the article is accepted for me with this version

Reviewer 2 Report

The Authors have addressed all of my concerns with the original manuscript.